# Synthesis, Cytotoxicity Assessment and Optical Properties Characterization of Colloidal GdPO_4_:Mn^2+^, Eu^3+^ for High Sensitivity Luminescent Nanothermometers Operating in the Physiological Temperature Range

**DOI:** 10.3390/nano10030421

**Published:** 2020-02-28

**Authors:** Kamila Maciejewska, Blazej Poźniak, Marta Tikhomirov, Adrianna Kobylińska, Łukasz Marciniak

**Affiliations:** 1Institute of Low Temperature and Structure Research, Polish Academy of Sciences, Okólna 2, 50-422 Wroclaw, Poland; akobylinska95@o2.pl; 2Department of Pharmacology and Toxicology, Faculty of Veterinary Medicine, Wrocław University of Environmental and Life Sciences, ul. C.K. Norwida 31, 50-366 Wrocław, Poland; blazej.pozniak@upwr.edu.pl (B.P.); marta.tikhomirov@upwr.edu.pl (M.T.)

**Keywords:** nanoparticles, synthesis, luminescent nanothermometers, transition metals, cytotoxicity, optical properties

## Abstract

Herein, a novel synthesis method of colloidal GdPO_4_:Mn^2+^,Eu^3+^ nanoparticles for luminescent nanothermometry is proposed. XRD, TEM, DLS, and zeta potential measurements confirmed the crystallographic purity and reproducible morphology of the obtained nanoparticles. The spectroscopic properties of GdPO_4_:Mn^2+^,Eu^3+^ with different amounts of Mn^2+^ and Eu^3+^ were analyzed in a physiological temperature range. It was found that GdPO_4_:1%Eu^3+^,10%Mn^2+^ nanoparticles revealed extraordinary performance for noncontact temperature sensing with relative sensitivity S_R_ = 8.88%/°C at 32 °C. Furthermore, the biocompatibility and safety of GdPO_4_:15%Mn^2+^,1%Eu^3+^ was confirmed by cytotoxicity studies. These results indicated that colloidal GdPO_4_ doped with Mn^2+^ and Eu^3+^ is a very promising candidate as a luminescent nanothermometer for in vitro applications.

## 1. Introduction

Nanotechnology is one of the most dynamically developing areas of contemporary science [1], which results, among others, from its importance in numerous biomedical applications. For example, nanosized inorganic particles enable performing noninvasive in situ diagnosis and therapy of biological systems. Light-induced drug release, or hyperthermia, magnetic and/or optical imaging, temperature, an pressure or pH noncontact readout are just a few examples that have become feasible thanks to the use of nanoparticles [2,3,4,5]. However, materials used for these purposes must meet several restrictive requirements before they can be applied, such as a lack of cytotoxicity, high chemical, thermal, and photo-stability, as well as high luminescence efficiency. The high stability of the colloidal solution of non-aggregated nanoparticles in aqueous media is another fundamental condition. For this purpose, the development of many sophisticated synthesis methods like thermal decomposition, polyol, hydro-, and solvo-thermal methods was a milestone in this field. The mentioned synthesis procedures enable creating multifunctional nanoparticles that combine excellent performance for both treatment and diagnosis. The precise in vivo treatment requires accurate and fast control of the physical and chemical parameters, among which temperature is one of the most fundamental ones. Therefore, noncontact thermal sensing techniques are of great importance.

Luminescent thermometry (LT) is one of the most promising techniques, which provides reliable performance of temperature readout in biological media. For this purpose, the temperature-dependent change of the spectral (position of the band, bandwidth, intensity or band shape) or temporal (rise or decay time) characteristics of nanophosphors is utilized for temperature readout [4,5,6,7,8,9,10,11,12,13,14]. In the case of biological applications, LT needs to reveal high sensitivity to temperature changes in the physiological temperature range (20–50 °C). Moreover, in order to enhance the accuracy and reliability of temperature sensing by the minimization of the light scattering and absorption by the tissue, emission of such phosphors should spectrally fall into biological optical windows (BWs) (BW-I: 650–950 nm, BW-II: 1000–1350 nm, and BW-III: 1500–1800 nm) [15,16,17,18,19,20,21]. Therefore, many potential non-invasive nanothermometers such as YAG garnets [22], NaYF_4_ fluorides [23], Au NPs [24], etc., doped with optical active ions operating in the range of BWs have been studied in the last few years. Nevertheless, these materials require difficult reaction conditions (e.g., high temperature, use of high boiling solvents) and further surface modifications to obtain biocompatible and stable nanoparticles in water solutions. That is why it is still desirable to seek for compounds that will be efficient and non-toxic phosphors and are obtained by inexpensive and simple synthesis procedures. One such synthetic techniques, classified as soft chemical routes, is the hydrothermal method [25]. Its main advantages are the use of water as a reaction environment and relatively low reaction temperature. What is more, many compounds obtained with this method do not require additional modifications to further obtain water-stable colloids. An example of such compounds are metal orthophosphates (MPO_4_, where M is metal), which are characterized by high stability in water. The distinctive features of MPO_4_ are their biocompatibility, high stability in physiological media, large index of refraction (n = 1.85), non-toxicity, and high thermal stability. [25,26,27,28]

Therefore, some examples of the use of orthophosphates doped with lanthanides for temperature sensing purposes have been already reported [29]. However, to the best of our knowledge, no attempt has been presented yet to use orthophosphates co-doped with transition metal (TM) and lanthanide (Ln) ions for temperature sensing. One of the most important advantages of using TM for this purpose is the strong dependence of their emission intensity on temperature changes. This results from the difference in the electronic configuration of the ground and excited states of the *d* orbital. The consequence of this fact is the intersection of potential energy parabolas of these states at *ΔE* energy (activation energy). Therefore, at elevated temperatures, the population of excited states is efficiently reduced by the nonradiative quenching. On the other hand, the lack of an intersection point between *4f* states of lanthanide Ln^3+^ ions results in their lower susceptibility to temperature quenching with respect to TM. Hence, their emission intensity may be utilized as a luminescent reference for ratiometric noncontact temperature sensors. Moreover, activation energy, in the case of TM, can be easily modified by the stoichiometry of the phosphor. So far, the use of thermometers based on the emission of transition metals such as Cr^3+^, V^3+,4+,5+^, Ti^3+,4+^, Co^2+^, Ni^2+^, and Mn^3+,4+^ has been reported in the literature [17,19,30,31,32,33]. Another interesting dopant is Mn and specifically Mn^2+^ ions.

The major advantage of using Mn^2+^ originates from strong coupling system due to the electron–phonon interaction in which luminescence is followed by the phenomenon of thermal quenching. Furthermore, integrating this ion with Eu^3+^ (where the weak coupling occurs in which the luminescence intensity is barely sensitive to temperature changes) in a single host material is permissive for further biomedical application. [34,35,36,37,38,39]. A highly sensitive luminescent thermometer based on Zn_2_SiO_4_:Mn^2+^-Gd_2_O_3_:Eu^3+^ nanocomposite was studied by Huang et al. [40]. In this work, relative sensitivity around 3%/°C at 30 °C was obtained. Although these results were very promising, the use of glass ceramics strongly hinders its applicability in the biomedical field. Therefore, using this pair of ions in a nanocrystalline matrix that is characterized as a stable in water colloid, with a size on the nanoscale, and negligible toxicity is a challenge that motivates our studies.

For that reason, in this article, we present a novel method that proceeds via a two-step synthesis procedure, namely precipitation and hydrothermal synthesis to obtain the colloidal GdPO_4_: Mn^2+^, Eu^3+^ nanoparticles. In addition, we propose to use these materials for a highly sensitive luminescent nanothermometer in the physiological temperature range. The highest recorded relative sensitivity for this material at 32 °C is 8.88%, which is, to the best of our knowledge, the highest S_R_ value of luminescent thermometers based on the emission of this pair of ions so far. Moreover, to prove the usefulness of this nanothermometer, cytotoxicity studies are conducted, which confirm the biosafety of this material.

## 2. Materials and Methods

### 2.1. Materials

The Mn^2+^, Eu^3+^-doped GdPO_4_ nanoparticles (NPs) were synthesized via the hydrothermal method. All chemicals: yttrium(III) oxide, Y_2_O_3_ (99.99%, Alfa Aesar, MA, USA), europium (III) oxide, Eu_2_O_3_ (99.9%, Alfa Aesar), manganese carbonate, MnCO_3_ (99.9%, Alfa Aesar, MA, USA), ammonium hydrogen phosphate, (NH_4_)_2_HPO_4_ (98.0%, Alfa Aesar, MA, USA), and urea (99.5%, Chempur, Poland), were used without further purification.

### 2.2. Synthesis

Appropriate amounts of oxides were diluted using a Teflon-lined autoclave in ultrapure nitrate acid to produce nitrates, followed by the evaporation of the excess of solution and drying over P_2_O_5_ in a vacuum desiccator for 1 day. The procedure of the synthesis of colloidal GdPO_4_: Mn^2+^, Eu^3+^ nanoparticles via the hydrothermal process contained two steps. The first step consisted of the precipitation of monodisperse hydroxycarbonates using an excess of urea and a precipitation temperature up to 90 °C. In the second step, the pH of the aqueous hydroxycarbonates slurry was reduced to pH = 4. The slightly acidic condition prevented the formation of a core-shell type material as in the case of the synthesis shown in [41]. Metal hydroxycarbonates were synthesized via homogeneous precipitation of lanthanide nitrates and manganese carbonate. The metal hydroxycarbonates were obtained by heating a water solution containing urea (3 mol/L) and metal precursors (5 × 10^−3^ mol/L) in a heating mantle at 90 °C under continuous stirring for 15 min. The concentration of Eu^3+^ (0.1%, 0.2%, 0.5%, 1%, 2%, and 5%) and Mn^2+^ (1%, 5%, 10%, and 15%) ions was changed with respect to Gd^3+^ ions (Appendix A.). After stirring, the suspension was then centrifuged. The resulting precipitate was washed three times with distilled water and dispersed in 30 cm^3^ of distilled water. Next, metal hydroxycarbonates were transferred to a Teflon bottle, with continuous stirring. The pH of the solution was lowered to 4 using nitric acid. Then, a saturated (NH_4_)_2_HPO_4_ water solution (0.17 mol/L) was added. After this time, the Teflon bottle was placed in the autoclave and kept at 200 °C for 2 h. After the reactor cooled down slowly, the resulting products of GdPO_4_: Mn^2+^, Eu^3+^ were washed and separated by centrifugation.

### 2.3. Methods

Powder diffraction studies were carried out using a PANalytical X’Pert Pro diffractometer equipped with an Anton Paar TCU 1000 N Temperature Control Unit using Ni-filtered Cu Ka radiation (V = 40 kV, I = 30 mA). Transmission electron microscopy images were obtained using a FEI Tecnai G2 20 X-TWIN microscope supplied with a CCD FEI Eagle 2K camera with a *High-angle annular dark-field* (HAADF) detector and electron gun with a LaB6 cathode.

The hydrodynamic size of the nanoparticles was determined by dynamic light scattering (DLS), conducted in a Malvern ZetaSizer at room temperature in a polystyrene cuvette, using distilled water as a dispersant. The content of the metals in the nanomaterials was determined by means of the ICP-OES technique, using the Thermo Scientific™ iCAP™ 7400 ICP-OES analyzer (Waltham, MA, USA). The emission spectra were measured using the 375 nm excitation line (laser diode) and the Silver-Nova Super Range TEC Spectrometer from Stellarnet (1 nm spectral resolution). The temperature of the sample was changed using a THMS 600 heating stage from Linkam (0.1 °C temperature stability and 0.1 °C set point resolution). The photoluminescence decay times were obtained using an FLS980 fluorescence spectrometer (Edinburgh Instruments, Livingston, UK) with an R928P side window photomultiplier tube as a detector (Hamamatsu, Japan) and micro-flash lamp as an excitation line measurement.

Cytotoxicity assessment was carried out on murine macrophage (J774.E) and human osteosarcoma (U2OS) cell lines. Cells were cultured in RPMI-1640 medium (Institute of Immunology and Experimental Therapy, Wrocław, Poland) supplemented with 10% fetal bovine serum (FBS, Sigma, Darmstadt, Germany), L-glutamine (Sigma, Darmstadt, Germany), and antibiotic (streptomycin and penicillin, Sigma, Germany). For the cytotoxicity assessment, cells were seeded in 96 well plates (TTP, Trasadingen, Switzerland) at a density of 3 × 10^3^ (U2OS) or 10 × 10^3^ (J774.E) cells per well and pre-incubated at 37 °C for 24 h in a humidified atmosphere of 5% CO_2_. After that, nanoparticle dispersions were added. Stock dispersions were prepared based on a simplified version of the NANOGENOTOX dispersion protocol. Nanoparticles were suspended in 0.05% BSA water solution and bath-sonicated at room temperature for up to 5 min. Next, the stock solutions were further diluted in 0.05% BSA, and dispersions in complete culture medium were prepared. In parallel, the sample with the highest nanoparticle concentrations were centrifuged at 30,000× *g* for 2 h, and the particle-free supernatants were used as a diluent control (to exclude any possible particle unrelated effects due to the presence of soluble, e.g., post-reaction, agents). Cells were exposed to the dispersions for 48 h (5% CO_2_, 37 °C). After that, the MTT assay was carried out. The test is based on the enzymatic reduction of the tetrazolium salt MTT (3-(4,5-dimethylthiazol-2-yl)-2,5-diphenyl-tetrazoliumbromide) in living, metabolically active cells. The metabolite, purple-colored formazan, was measured colorimetrically, using a multiwell plate reader. A preliminary experiment showed no interference of the nanoparticles with MTT in a cell-free system. After 4 h of incubation at 37 °C, 80 μL of lysis buffer were added. The optical density (OD) was measured after 24 h using a spectrophotometric microplate reader (Tecan Spark 10M, Switzerland) at a wavelength of 570 nm (reference 630 nm). The OD of control cells was taken as 100%. Cell viability was determined as follows: % viability = (mean OD in the test wells/mean OD for control wells) × 100. Where possible, the half maximal inhibitory concentration (IC_50_) was calculated. The results were obtained from at least 3 independent experiments.

## 3. Results and Discussion

### 3.1. Structural and Morphological Properties of GdPO_4_:Mn^2+^, Eu^3+^ Nanoparticles

Pure phase hexagonal GdPO_4_: Mn^2+^, Eu^3+^ nanoparticles were obtained, which crystallized in the hexagonal rhabdophane-type GdPO_4_ hydrate (GdPO_4_·nH_2_O) P6222 space group. All of the diffraction patterns obtained for the nanoparticles with different concentrations of Eu^3+^ (Figure 1a) corresponded to the reference LaPO_4_ (ICSD-31654) pattern, which pointed out that the dopant concentration did not introduce any impurity phases. Analogously for nanoparticles with a different concentration of Mn^2+^ (verified by the ICP measurements; see Appendix A), there were no additional peaks in the diffractograms (see also Appendix A). However, XRD peaks shifted toward smaller angles with rising Mn^2+^ concentration. The detailed analysis of the diffraction patterns using the Rietveld refinement technique revealed that this effect was associated with the enlargement of the crystallographic cell in the *c* direction from 6.14 Å (1% Mn^2+^) to 6.3 Å (15% Mn^2+^), along with the (PO_4_)^3−^ tetrahedral chains (Figure 1b). It is worth noting here that for 30% of Mn^2+^ concentration, the crystallographic structure of GdPO_4_ did not remain stable, and the separation of additional crystallographic phases was observed. Therefore, in these studies, 15% of Mn^2+^ was used as the highest allowed dopant concentration. The ionic radii of Mn^2+^ ions (0.96 Å) in the dodecahedral site were smaller than the Gd^3+^ one (1.053 Å); however, substitution of Gd^3+^ ions by the Mn^2+^ led to the distortion of the dodecahedral, despite that the ionic radii of Mn^2+^ ions (0.96 Å) in the dodecahedral site were slightly smaller than the Gd^3+^ ones (1.053 Å) [42]. This was related to the uncompensated charge between these ions, which led to the generation of the defect states. A similar observation was previously reported by Nono [43]. On the other hand, Eu^3+^ concentration did not affect the position of the diffraction peaks.

The hydrodynamic radius of the nanoparticles, as measured using the DLS technique, was around 60 nm (Figure 1c) and was independent of dopant concentration. The high stability of the colloidal solution of GdPO_4_:Mn^2+^, Eu^3+^ nanoparticles in water was confirmed by the zeta potential measurement (−45 mV) (Figure 1d). However, the analysis of representative TEM images (Figure 1e) revealed that the obtained phosphor was composed of rod-shaped nanoparticles of an average length around 48 nm (Figure 1c) and a width around 7 nm.

### 3.2. Spectroscopic Properties of GdPO_4_:Mn^2+^, Eu^3+^ Nanoparticles

The energy level diagram (Figure 2a) indicated that for Mn^2+^ ions in O_h_ site symmetry with the [^4^T_1_ (^4^G)→^6^A_1_ (^6^S)] transition, the electronic transition was expected at about 550 nm. However, the actual site symmetry of Gd^3+^ was D_2_, and therefore, the ^4^T_1_ level split into ^2^A_1_, ^4^B_2_, and ^4^B_3_; as a result, an inhomogeneously broadened emission band of Mn^2+^ was observed in the emission spectrum (Figure 2b). Besides the Mn^2+^ emission band, four narrow lines at 590, 620, 650, and 700 nm, which could be attributed to transitions from the ^5^D_0_ to ^7^F_1_, ^7^F_2_, ^7^F_3_, and ^7^F_4_ levels of Eu^3+^ ions, respectively, could be observed. In the excitation spectrum (Figure 2b) recorded when monitoring emission at 550 nm, a narrow sharp band of Mn^2+^ at 405 nm was observed, which could be attributed to the ^6^A_1_ (^6^S) → ^4^A_1_ (^4^G), ^4^E (^4^G) transition. On the other hand, the excitation spectrum monitored at 695 nm contained sharp bands associated with Laporte-forbidden *f-f* transitions of Eu^3+^ ions located at 361, 383, 392, and 415 nm, corresponding to ^7^F_1_→^5^D_4_, ^7^F_0_→^5^L_7_, ^7^F_0_→^5^L_6_, ^7^F_0_→^5^D_3_ transitions, respectively (Figure 2b; see also Appendix A).

The fact that no absorption band of Eu^3+^ occurred in the excitation spectra of Mn^2+^ (λ_em_ = 550 nm) confirmed the lack of nonradiative energy transfer between Mn^2+^ and Eu^3+^ ions. This was especially important information, since the population of the Ln^3+^ excited state via TM→Ln^3+^ energy transfer may efficiently hinder the relative sensitivity of the luminescent thermometer to temperature changes [16]. Moreover, the luminescent properties of TM ions are especially prone to modification of the ions’ local symmetry. The calculation of cell parameters using the Rietveld refinement method revealed Mn^2+^ concentration-dependent elongation of the *c* lattice constant. Therefore, in order to verify the influence of the dopant concentration on the susceptibility of the emission intensities of both ions to thermal quenching, nanoparticles with different admixtures of Mn^2+^ and Eu^3+^ ions were synthesized.

The comparison of the room temperature emission spectra of GdPO_4_ nanoparticles doped with 0.1, 0.2, 0.5, 1, 2, and 5% Eu^3+^ and 10% Mn^2+^ revealed that the increase of Eu^3+^ concentration enhanced the emission intensity of Eu^3+^, which could be explained in terms of the growing number of emitting centers (Figure 2c). In the case of the emission spectra of GdPO_4_: 1%Eu^3+^, x%Mn^2+^ measured as a function of Mn^2+^ concentration, the highest emission intensity of the ^4^T_1_ (^4^G)→^6^A_1_ (^6^S) electronic transition was found for nanoparticles doped with 5% of Mn^2+^, then the increase of dopant concentration (10, 15%) led to the gradual quenching of the ^4^T_1_ (^4^G)→^6^A_1_ (^6^S) emission intensity. Analogous changes of the emission spectra could be found in the case of the tuned Mn^2+^ concentration (Appendix A). In order to quantify the observed changes, Figure 2d and e show diagrams representing the contribution of the particular ion emission intensity normalized to the total emission spectra. Initially, at a low Eu^3+^ concentration, the Mn^2+^ emission band dominated in the spectra. However, the increase of Eu^3+^ concentration caused gradual enhancement of its emission intensity, and above 2%, Eu^3+^ emission became dominant. Analogously, enlargement of Mn^2+^ concentration led to the enhancement of the ^4^T_1_ (^4^G)→^6^A_1_ (^6^S) band intensity with respect to the ^5^D_0_→^7^F_J_ bands. The observed changes were manifested as a gradual modification of the emission color from greenish for 10% Mn^2+^, 0.1%Eu^3+^ to orange-red for 10% Mn^2+^, 5% Eu^3+^ (Figure 2f).

### 3.3. Kinetics of Emission of Eu^3+^ and Mn^2+^ in GdPO_4_ Nanoparticles

In order to provide deeper insight into and understanding of how the concentration of dopant ions affected the spectroscopic properties of GdPO_4_: Mn^2+^, Eu^3+^ nanoparticles, the kinetics of the excited state of Eu^3+^ (^5^D_0_→^7^F_4_ transition, λ_em_ =695 nm) and Mn^2+^ (^4^T_1_ (^4^G)→^6^A_1_ (^6^S) transition, λ_em_ =550 nm) ions were investigated (Figure 3). Due to the non-exponential shape of the obtained curves (I(t)), the average decay times were calculated according to the following formula:〈τ〉=∫I(t)tdt/∫I(t)dt. For nanoparticles singly doped with Mn^2+^ (10%) or Eu^3+^ (1%) ions, the average luminescence decay was equal to 12.85 ms and 1.69 ms, respectively. As can be seen in Figure 3c, Eu^3+^ concentration had some influence on the lifetime of the ^5^D_0_ state. The calculation of the average decay time revealed that its value shortened from 1.69 ms to 1.37 ms. Additionally, its decay profile became non-exponential. This effect may be explained in terms of energy diffusion among the excited state of Eu^3+^ ions followed by the transfer to the surface quenching centers. The average distance between Eu^3+^ ions shortened with the increase of dopant concentration, which facilitated the probability of energy diffusion. Similar observations were made by Yaiphaba et al. for core and core/shell GdPO_4_:Eu^3+^ nanoparticles. Moreover, the increase of the Eu^3+^ concentration due to the difference in the ionic radii between Gd^3+^ and Eu^3+^ may lead to the change of the site local symmetry. In order to confirm this hypothesis, a series of nanoparticles singly doped with different concentrations of Eu^3+^ ions was synthesized. It is well known that the emission intensity ratio of ^5^D_0_→^7^F_2_ (electric dipole transition) increases with respect to ^5^D_0_→^7^F_1_ (magnetic dipole transition) when the disorder of the ions’ local environment increases. Therefore, their emission intensity ratio is often used as a luminescent probe of ions’ local symmetry changes (disorder parameter). In the case of GdPO_4_:Eu^3+^ nanoparticles, the disorder parameter increased from 1.4 for 0.1% Eu^3+^ ions to 1.87 for 5%Eu^3+^ ions (Appendix A). This confirmed that in our case, some lowering of the local symmetry of Eu^3+^ ions occurred when the concentration of Eu^3+^ increased.

On the other hand, in the case of the Mn^2+^ luminescence decay profile, apparent shortening of the lifetime was observed along with an increase of Eu^3+^ ion concentration. The in-depth analysis indicated that with increase of the Eu^3+^ concentration, additional short-leaving components appeared in the decay profile. The double exponential fit of Mn^2+^ decay profiles yielded the same value of the time constant (τ_2_) for the longer component of the decay time, which clearly indicated that no concentration effect was observed (Appendix A). The average lifetime calculated for the 10–50 ms range did not change with Eu^3+^ concentration (Figure 3c). The double exponential fit of Mn^2+^ decay profiles clearly indicated that no concentration dependence was observed (Appendix A).

The increase of Mn^2+^ concentration led to the shortening of both the Mn^2+^ (from 12.25 ms for 1%Mn^2+^ to 11.6 ms for 10%) and Eu^3+^ (from 1.6 ms for 0.1%Eu^3+^ to 1.35 ms for 5%) decay times. The shortening of Mn^2+^ with the increase of its concentration was explained by many authors in terms of the Mn^2+^→Eu^3+^ energy transfer [44]. However, in each of the analyzed systems, this energy transfer was additionally confirmed by the occurrences of the absorption bands of Eu^3+^ ion in the excitation spectrum of Mn^2+^ and vice versa. In our case, such energy transfer was excluded by the excitation spectra measurements (Appendix A).

Although, the shortening of the Eu^3+^ decay time with Mn^2+^ concentration was observed, it revealed a similar tendency to that measured for Mn^2+^, which confirmed that the modification of the ions’ local symmetry with the increase of Mn^2+^ concentration was responsible for the observed lifetime changes. The change of ions’ local symmetry was found when Eu^3+^ concentration increased. Therefore, even a more spectacular effect was expected for Mn^2+^ concentration series, because the difference of the ionic radii between Gd^3+^ and Mn^2+^ was larger.

### 3.4. Luminescent Thermometry

The fact that there was no energy transfer between dopants is especially desirable for luminescent nanothermometry purposes, because such a quenching channel may significantly hinder the susceptibility of nanothermometer emission to temperature changes. The thermal measurement of the emission intensity of nanoparticles singly doped with 1%Eu^3+^ (Appendix A) indicated that the emission intensity of Eu^3+^ was almost temperature independent in the whole analyzed temperature range. This confirmed that Eu^3+^ luminescence could be treated as an appropriate luminescent reference. This result was in agreement with our expectations, because the emitting ^5^D_0_ state was separated by around 11,500 cm^−1^ from the first lower laying state (^7^F_6_). This energy gap sufficiently minimized the probability of its depopulation via multiphonon processes. Additionally, the cross-section of *f* states with CT (charge transfer) parabola, which is frequently considered as a main mechanism responsible for thermal quenching of Eu^3+^ luminescence, was located at an energy that was significantly higher than the provided thermal energies in the temperature range under consideration. On the other hand, the Mn^2+^ emission was susceptible to thermal quenching (Appendix A). This was associated with the occurrence of the intersection point between ground and excited states at Ea= 875 cm^−1^ (Appendix A). The relatively small value of activation energy was in agreement with the observed strong thermal quenching of its emission intensity (Appendix A). Consequently, to assess the performance of luminescence nanothermometers based on the optical response of GdPO_4_: (x%)Mn^2+^, (y%)Eu^3+^ for temperature readout, temperature measurements were carried out. In the case of GdPO_4_: (1, 5, 10, 15%)Mn^2+^, (1%)Eu^3+^ and GdPO_4_: (10%)Mn^2+^, (0.1, 0.2, 0.5, 1, 2, 5%)Eu^3+^ nanoparticles, the increase of the temperature led to the quenching of both Mn^2+^ and Eu^3+^ emission (Appendix A). However, the quenching rate of particular ions differed. For a constant concentration of Eu^3+^, it could be found that the increase of temperature led to the quenching of both the Mn^2+^ and Eu^3+^ emission intensities (Appendix A). For a constant concentration of Eu^3+^ ions, the thermal quenching rate of the emission intensity of Mn^2+^ gradually increased with Mn^2+^ concentration. A similar dependence was observed for Eu^3+^ emission intensity, which was, however, less susceptible to temperature changes for a Mn^2+^ concentration below 10%. For 15% of Mn^2+^, the Eu^3+^ emission intensity revealed similar quenching dependence as Mn^2+^ emission intensity. In order to utilize the observed differences in the thermal quenching rate between the ^4^T_1_ (^4^G)→^6^A_1_ (^6^S) electronic transition of Mn^2+^ ions and the ^5^D_0_→^7^F_4_ electronic transition of Eu^3+^ ions, their emission intensity ratio (LIR) could be used as an accurate thermometric parameter. Since emission of Mn^2+^ and Eu^3+^ partially overlapped, the integral emission intensity of ^5^D_0_→^7^F_4_ electronic transition, which to the least extant overlapped with the Mn^2+^ band, was taken as representative for Eu^3+^ emission intensity in the performed studies to minimize the inaccuracy in the intensity determination. Accordingly, the LIR is defined as:(1)$$\mathrm{LIR}=\frac{\;I\;\left({\mathrm{Mn}}^{2+}\right)}{I\;\left({\mathrm{Eu}}^{3+}\right)}=\frac{\int I\left({}^{4}T_{1}{(}^{4}G)\to^{6}A_{1}\left({}^{6}S\right)\right)}{\int I\left({}^{5}D_{0}\to^{7}F_{4}\right)}$$

Taking advantage of the calculated integral emission intensities of both electronic transitions as a function of temperature (Appendix A), the thermal dependence of LIR was analyzed as a function of Mn^2+^ concentration (Appendix A). For a low Mn^2+^ dopant amount, LIR decreased initially by around 10% and above 40 °C started to saturate. However, the increase of Mn^2+^ concentration resulted in the gradual enhancement of the thermal decrease of LIR up to 10% of Mn^2+^ ions for which a decrease by 60% in the temperature range under consideration was found. Above that concentration, the observed changes of LIR were less significant. In order to quantify the observed changes and to verify the utility of LTs based on GdPO_4_: (x%)Mn^2+^, (y%)Eu^3+^ emission to temperature readout, their relative sensitivity was calculated according to the following formula:(2)$$S_{R_{}}=\frac{\;1}{\mathrm{LIR}}\frac{\Delta \mathrm{LIR}}{\Delta T}100\%$$
where ΔLIR represents the change of the LIR’s value corresponding to a ΔT change of temperatures. Evidently for most of Mn^2+^ concentrations, S_R_ decreased with the increase of the temperature. Strong changes of LIR as a function of temperature observed for GdPO_4_:10% Mn^2+^, 1% Eu^3+^ had a reflection in relative sensitivity, which reached 8.88%/°C at 32 °C and gradually decreased to around 4%/°C at temperatures above 38 °C. (Appendix A). Therefore, a 10% Mn^2+^ concentration was chosen for optimization of Eu^3+^ ions’ concentration (Figure 4). However, absolute emission intensity also decreased with rising Mn^2+^ concentration. Similar observations were made for Eu^3+^ emission intensity. Thermal evolution of emission spectra of GdPO_4_: 10% Mn^2+^, x% Eu^3+^ nanoparticles with different concentrations of Eu^3+^ measured in the 30–50 °C temperature range are presented in Appendix A. It can be seen that, for a low dopant concentration of Eu^3+^ ions, the emission intensity of Eu^3+^ was barely dependent on temperature. However, when the Eu^3+^ concentration increased, the quenching of Eu^3+^ emission was more pronounced and for nanoparticles co-doped with 5%Eu^3+^ and 10% Mn^2+^ emission intensity at 50 °C reached 86% of the initial (at 30 °C) value.

This effect was probably associated with the energy diffusion among Eu^3+^ excited state and then the luminescence quenching centers, i.e., surface defects. As was shown, the rod-like GdPO_4_ nanoparticles were around 7 nm in diameter, and therefore, energy diffusion to the surface defects was especially probable. Moreover, the Eu^3+^ site symmetry decreased with increasing of Eu^3+^ ion concentration, what facilitated luminescence thermal quenching.

The Mn^2+^ emission intensity was reduced by 20–25% in the analyzed temperature range, even for nanocrystals with a low Eu^3+^ concentration; however, the strongest changes were observed for 1% of Eu^3+^. Clearly, the maximal value of S_R_ was a function of dopant concentration. Therefore, in Figure 4d, S_R_ at 32 °C is plotted against Mn^2+^ concentration. The maximal value changed as follows 1.65%/°C for 1% Mn^2+^; 3.39 %/°C for 5%Mn^2+^; 8.88 %/°C for 10% Mn^2+^, and 0.005%/°C for 15% Mn^2+^. The high relative sensitivity of GdPO_4_: Mn^2+^, Eu^3+^ resulted from the high susceptibility of Mn^2+^ emission intensity to temperature changes. As was already shown for a higher Mn^2+^ concentration, the ions’ site symmetry decreased, facilitating thermal quenching of the Mn^2+^ emission intensity. Hence, S_R_ rose. However, for 15% Mn^2+^, probably due to the large change of the ions’ local symmetry, the profile of the thermal quenching of Eu^3+^ emission intensity revealed a similar shape to the one observed for Mn^2+^. Therefore, LIR changed only slightly with temperature.

In the case of nanoparticles with different Eu^3+^ concentrations, the lowest value of the relative sensitivity was observed for 0.2% Eu^3+^ (S_R_ = 0.24%/°C), and the maximal value of S_R_=8.88%/°C was obtained for 1% of Eu^3+^. Above this dopant concentration, relative sensitivity decreased to S_R_=7.03%/°C for 2% Eu^3+^ and 4.53%/°C for 5% Eu^3+^. Taking advantage of these studies, it could be concluded that GdPO_4_: 1% Eu^3+^, 10% Mn^2+^ nanoparticles revealed the best performance for temperature sensing. The measurement of the LIR’s value for GdPO_4_:10% Mn^2+^, 1% Eu^3+^ nanoparticles within the four heating-cooling cycles confirmed the high thermal stability of the synthesized nanoparticles and the reversibility of observed the thermal LIR changes. The comparison of S_R_ for different Mn-based luminescent thermometers (Table 1) confirmed the high applicative potential of the studied GdPO_4_:Eu^3+^,Mn^2+^ nanoparticles.

Due to the fact that all colloidal GdPO_4_ doped with different amounts of Mn^2+^ and Eu^3+^ ions showed high sensitivity in the physiological range, we decided to investigate the cytotoxic response to the obtained material in vitro. The cytotoxicity of GdPO_4_ nanoparticles doped with lanthanide ions has been already assessed, confirming their good performance for biological application [49]. Therefore, the main cytotoxic aspect of nanoparticles investigated here may be related to the presence of Mn^2+^ ions. Hence, we decided to perform a cytotoxicity assessment using the highest Mn^2+^ concentration under investigation (GdPO_4_: 15%Mn^2+^, 1%Eu^3+^ nanoparticles). High biocompatibility is necessary to consider biological applications of any material. The choice of the in vitro model was based on the fact that under in vivo conditions, macrophages form the primary line of response to particulate matter. Thus, they are responsible for the distribution and clearance of nanoparticles and their agglomerates. On the other hand, U2OS is a cancer cell line derived from bone tissue, which is a common target tissue for different ceramic biomaterials. The effects of GdPO_4_: 15%Mn^2+^, 1%Eu^3+^ nanoparticles on cell viability are summarized in Figure 5. In the case of GdPO_4_: 15%Mn^2+^, 1%Eu^3+^ nanoparticles, no significant effect on cell viability was observed, even at the highest concentration of 100 µg/mL. Moreover, a slight stimulatory effect was observed in all dilutions in U2OS cells. The nanoparticle concentrations seemed to have no clear relation to this phenomenon. The possible explanation to this fact may be the dye-particle interactions [50], changes in numerous enzymes, energy homeostasis, or oxidative stress [41]. Although a preliminary experiment showed no interaction between the particles and MTT in an acellular system, the other potential factors could not be entirely excluded [51]. Nevertheless, this effect was negligible.

In conclusion, GdPO_4_: 15%Mn^2+^, 1%Eu^3+^ was found to be biocompatible in the applied in vitro model. The excitation wavelength used in these studies prevented the in vivo temperature readout. However, sensitization of Mn^2+^ via the up-conversion process may be a solution for this limitation. Therefore, this approach should be verified in further investigations.

## 4. Conclusions

In this work, a novel synthesis method of colloidal GdPO_4_:Mn^2+^, Eu^3+^ doped with different amounts of Mn^2+^ and Eu^3+^ nanoparticles was described. The synthesis method contained two steps: (i) the precipitation of metal hydroxycarbonates and (ii) hydrothermal synthesis in a slightly acid condition. XRD patterns confirmed highly purified single phase materials even with a high amount of the dopants. Furthermore, the synthesized nanoparticles revealed a rod-like morphology of about 48 nm in length and 7 nm in width (TEM). Moreover, high stability in water solution was confirmed by zeta potential measurements (−45 mV). Furthermore, the spectroscopic properties of GdPO_4_ doped with different concentrations of Eu^3+^ and Mn^2+^ ions were investigated. It was found that the increase of Eu^3+^ concentration led to lowering of the ions’ local symmetry. As a consequence, shortening of the ^5^D_0_ luminescence decay and the enhancement of the Eu^3+^ luminescence thermal quenching were observed. More spectacular changes were found for changed Mn^2+^ dopant concentration due to large difference in the ionic radii and charge between Mn^2+^ and Gd^3+^. Therefore, the luminescence thermal quenching rate increased with Mn^2+^ concentration. Because the energy transfer between Eu^3+^ and Mn^2+^ ions was excluded by the excitation spectra measurement, the observed modification of the emission intensity ratio could be explained in terms of the dopant concentration-induced change of the ions’ local symmetry. Consequently, the difference in the luminescence thermal quenching between these ions and their luminescence intensity ratio could be used for noncontact temperature sensing. The best performance for temperature sensing revealed GdPO_4_: 10%Mn^2+^, 1%Eu^3+^ nanoparticles with a relative sensitivity S_R_=8.88%/°C at 32 °C. Furthermore, the cytotoxicity assessment excluded the toxicity of colloidal GdPO_4_: Mn^2+^, Eu^3+^. Summarizing, the colloidal GdPO_4_: Mn^2+^, Eu^3+^ nanoparticles that were highly stable in water solution were obtained through a novel synthesis method. The obtained nanoparticles exhibited negligible cytotoxicity, and for this reason, they show promise for in vitro applications of luminescent nanothermometry. Moreover, the presented nanoparticles with different amounts of Mn^2+^ and Eu^3+^ showed a high relative sensitivity (S_R_) value in a physiological temperature range.

## Figures and Tables

**Figure 1 nanomaterials-10-00421-f001:**
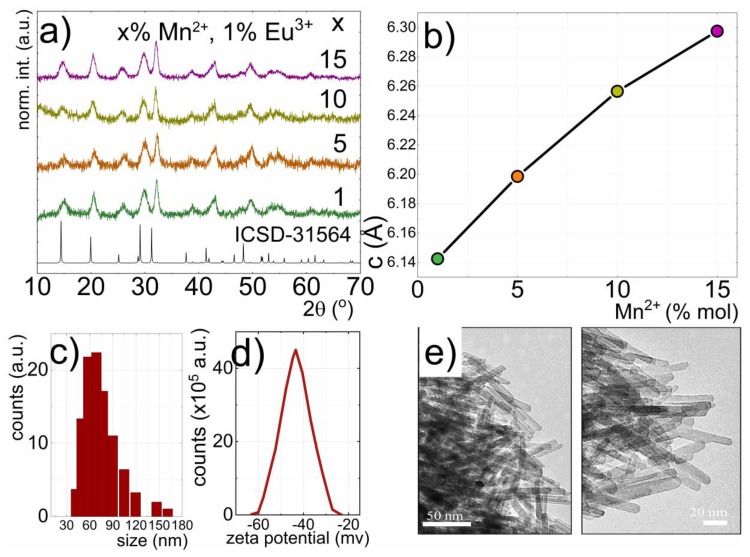
XRD patterns for GdPO_4_ doped with 1, 5, 10, and 15% Mn^2+^; 1% Eu^3+^ (**a**); influence of the Mn^2+^ concentration on the *c* cell parameter (**b**); the GdPO_4_: 10% Mn^2+^, 1% Eu^3+^ nanoparticle size distribution determined from DLS measurement (**c**); zeta potential of their colloidal solution in water (**d**); the representative TEM images of GdPO_4_: 10% Mn^2+^, 1% Eu^3+^ nanoparticles (**e**).

**Figure 2 nanomaterials-10-00421-f002:**
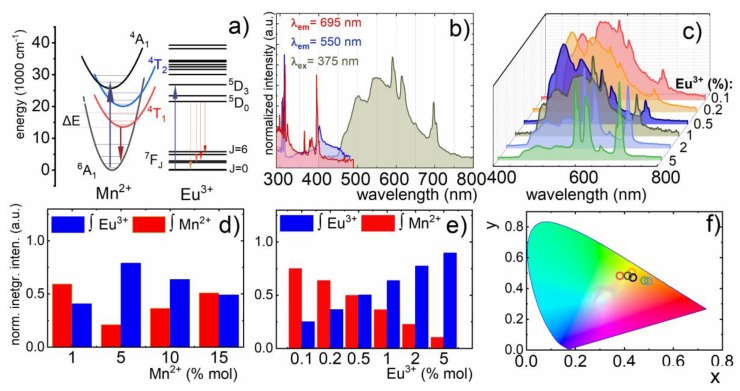
Schematic energy level diagram of Mn^2+^ and Eu^3+^ ions (**a**); the comparison of the excitation spectra for *λ*_em_ = 550 nm and *λ*_em_ = 695 nm and emission spectra *λ*_ex_ = 375 nm measured at 30 °C of GdPO_4_: 10%Mn^2+^, 1%Eu^3+^ nanoparticles (**b**); the comparison of the emission spectra of GdPO_4_ with 0.2, 0.5, 1, 2, and 5% Eu^3+^ and 10%Mn^2+^ concentration measured upon 375 nm excitation (**c**); contribution of the particular ion’s emission intensity to the total emission spectra for different Mn^2+^ (**d**) and Eu^3+^ (**e**) concentrations at 30 °C; and the CIE diagram for GdPO_4_: 10% Mn^2+^, x% Eu^3+^ (x=0.1, 0.2, 0.5, 1, 2, 5%) (**f**).

**Figure 3 nanomaterials-10-00421-f003:**
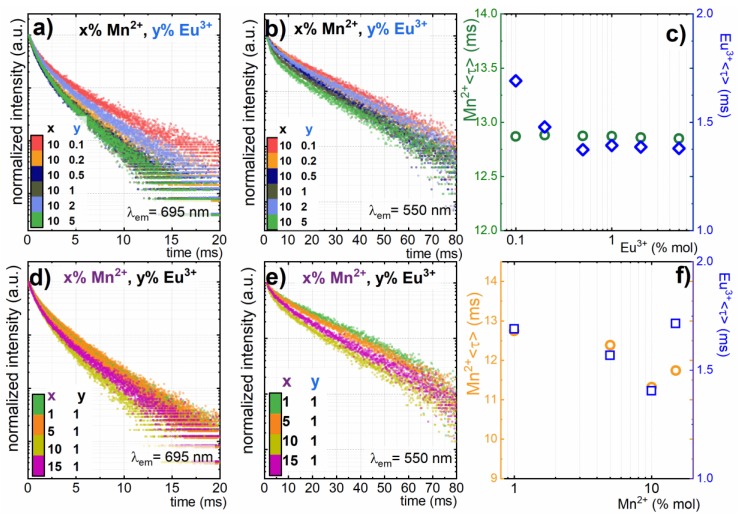
Luminescence decay curves for GdPO_4_: 10%Mn^2+^, 1%Eu^3+^ nanoparticles recorded at λ_em_=550 (**b**,**e**) (^4^T_1_ (^4^G)→^6^A_1_ (^6^S) electronic transition of Mn^2+^) and λ_em_=695 nm (**a**,**d**) (^5^D_0_→
^7^F_4_ electronic transition of Eu^3+^) and average decay times of GdPO_4_: 10%Mn^2+^, y%Eu^3+^ (**c**) and GdPO_4_: x%Mn^2+^, 1%Eu^3+^ (**f**) nanoparticles doped with different concentrations of Mn^2+^ and Eu^3+^ ions.

**Figure 4 nanomaterials-10-00421-f004:**
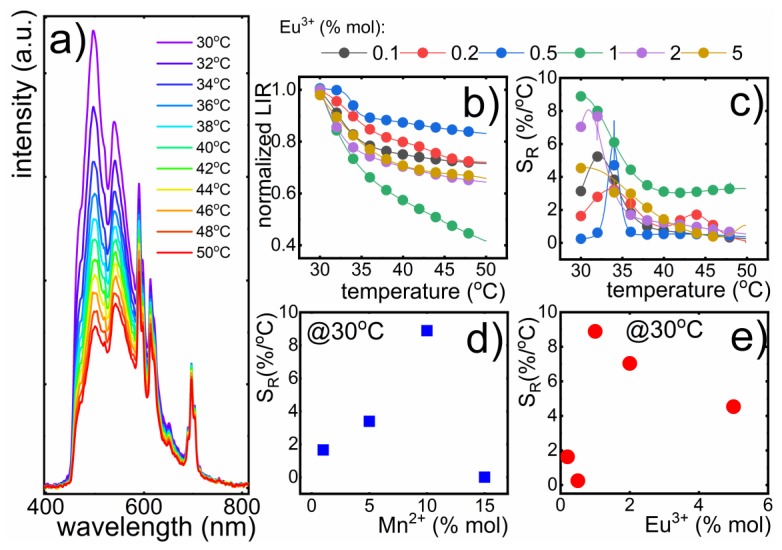
Thermal evolution of the emission spectra of GdPO_4_^:^ 1%Eu^3+^, 10%Mn^2+^ nanoparticles (**a**); LIR (**b**) and S_R_ (**c**) as a function of temperature for different Eu^3+^ concentration; the maximal values of S_R_ for different concentration of Mn^2+^ (1, 5, 10, 15%) with 1% Eu^3+^ ions (**d**) and Eu^3+^ (0.1, 0.2, 0.5, 1, 2, 5%) with 10% Mn^2+^ions (**e**).

**Figure 5 nanomaterials-10-00421-f005:**
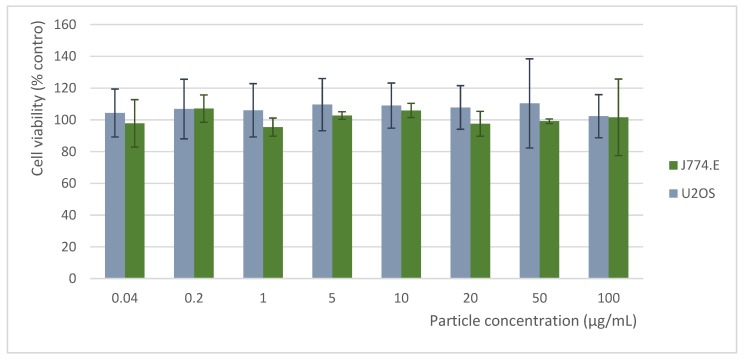
Mean (±SD) viability of J774.E murine macrophages and U2OS human osteosarcoma cells exposed for 48 h to different concentrations of GdPO_4_: 15%Mn^2+^, 1%Eu^3+^. Viability determined by MTT assay and expressed as the percent of control (results obtained from three independent experiments).

**Table 1 nanomaterials-10-00421-t001:** Comparison of the parameters for different manganese-based luminescent thermometers.

Compound	Mn Valence State	Temperature Range (°C)	S_R_ Max (%/°C)	Reference
Y_3_Al_5_O_12_:Mn^3+^, Mn^4+^, Nd^3+^	III/IV	−90–523	2.69	[17]
ZnGeO_4_:Mn^2+^	II	250–420	12.2	[35]
MgTiO_3_:Mn^4+^ a	IV	−200–50	1.2	[45]
Mn^2+^:Zn_2_SiO_4_–Eu^3+^:Gd_2_O_3_	II	30–50	3.05	[40]
Zn_2_SiO_4_:Mn^2+^	II	0–300	12.2	[46]
Eu^3+^/Mn^4+^:YAG	IV	20–120	4.81	[47]
Tb^3+^/Mn^4+^:YAG	IV	20–120	3.73	[47]
CsPb(Cl/Br)_3_:Mn^2+^	II	−193–20	10.04	[48]
GdPO_4_:10%Mn^2+^, 1%Eu^3+^	II	30–50	8.88	This work

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
