# Peer review of "Synthesis, Cytotoxicity Assessment and Optical Properties Characterization of Colloidal GdPO4:Mn2+, Eu3+ for High Sensitivity Luminescent Nanothermometers Operating in the Physiological Temperature Range"

_nanomaterials, 2020, doi:10.3390/nano10030421_

Round 1
Reviewer 1 Report
The manuscript entitled “Synthesis, cytotoxicity assessment and optical properties characterization of colloidal GdPO4: Mn2+, Eu3+ for high sensitivity luminescent nanothermometers operating in the physiological temperature range” describe synthesis, luminescence, toxicity and thermometry properties of colloidal GdPO4: Mn2+, Eu3+ nanoparticles. The main application sought is for in vitro/in vivo luminescence thermometry. The maximum relative sensitivity achieved is relatively appealing, around 9 %/°C at 32 °C for the optimised concentrations of the two dopants. The manuscript may be published in Nanomaterials after major revision.
Major comments
1. lease, explain the readers why the system described may be suitable for in vitro/in vivo nanothermometry since the excitation wavelength at 375 nm is not biologically friendly.
2. The text describing the potential occurrence of (bidirectional) energy transfer between Mn and Eu is confusing and difficult to follow. The donor -acceptor scheme should be applied to both Mn and Eu with the two spectral evidences for a potential energy transfer (excitation spectra and the evolution of the emission decays of the donor (Mn, Eu) at fixed concentration with increasing concentration of the acceptor (Eu, Mn). Although both evidences are partially discussed by the authors, the text should be shortened and more systematised.
3. In case of Mn2+ , aliovalent doping induces defects via charge-compensation, (probably, oxygen vacancies or other type ?). Though authors discuss the appearance of the so-called "defect states" this is not equivalent with the generation of defects, whatever their nature. Please, give a short description of the defects generated whose amount is likely to be significant for 10% Mn2+. Their distribution as near-neighbour or/and next near- neighbour to Eu is expected to affect drastically the Eu luminescence (emission shape and dynamics).
Minor comments:
Some references are incomplete: e.g. [2 - 18, 21, 24 - 28, 31, 32, 34 - 39, 41 - 43, 45 - 52] (it lacks the title of the article, Journal, year, vol, issue and pages). Please, correct them. Also, in Introduction, row 49 “non-invasive nanothermometers such as YAG garnets” should probably cite Ref 15 and 20 not22. In ref 22 the study is on LiLaP4O12 nanocrystals. Further, the authors should cite Revised effective ionic radii and systematic studies of interatomic distances in halides and chalcogenides. Acta Crystallographica Section A: Crystal Physics, Diffraction, Theoretical and General Crystallography 1976, 32 (5), 751-767) when discussing the ionic radii of the cation/dopants. Apparently, the XRD patterns do not match closely those of PDF of LaPO4 (Figure 1a). Probably, it would be better of authors repeat the XRD measurements. Further, the TEM images in Figure 1e displays some agglomeration contrary to the authors description. Please revise the text accordingly. Row 200 “narrow sharp band at 405 nm is observed”. Here the authors describe the excitation spectra monitoring the 550 nm Mn2+ emission although in the Figure2b it can be observed that the excitation spectra (the blue lined spectra) displays the absorption peak at ~450 nm. Please verify the peak wavelength. What is the significance of the absorption peaks at 300 – 325 nm ? How the authors extract the contribution of the particular ion’s emission intensity from the total emission spectra (spectral deconvolution ?) It appears that Table 1, is not cited or discussed in the manuscript. Please discus the thermometers performance in comparison with the reported literature. In Figure S4b, the asymmetry ratio value of the 0.5%Eu sample is out of trend. Can the authors explain this ? . Please , /complete the caption of Figure S10 (temperature range). Further, Figure S8, the “Thermal evolution of integrated Eu3+” has an out of trend evolution for 0.5%Eu which needs some explanation. In Figure S9, the LIR evolution of 5% and 10% Eu is similar for the first two temperature values (30 and 32 °C), but the corresponding Sr are quite different (8.88%/°C versus ~4%/°C). The authors should verify the LIR and Sr values. Some of the Figure and Figure captions in Supplemental Information are mislabelled with Mn3+ instead of Mn2+. Please correct the English grammar and language through the text (local ions symmetry should be replaced with ions local symmetry and so on). Also, several verbs are missing.
Author Response
The manuscript entitled “Synthesis, cytotoxicity assessment and optical properties characterization of colloidal GdPO4: Mn2+, Eu3+ for high sensitivity luminescent nanothermometers operating in the physiological temperature range” describe synthesis, luminescence, toxicity and thermometry properties of colloidal GdPO4: Mn2+, Eu3+ nanoparticles. The main application sought is for in vitro/in vivo luminescence thermometry. The maximum relative sensitivity achieved is relatively appealing, around 9 %/°C at 32 °C for the optimised concentrations of the two dopants. The manuscript may be published in Nanomaterials after major revision.
Major comments
- lease, explain the readers why the system described may be suitable for in vitro/in vivo nanothermometry since the excitation wavelength at 375 nm is not biologically friendly.
Author's replay:
Thank you very much for this comment. We agree with the suggestion of the Reviewer that UV/blue excitation is not appropriate for in vivo application. However, most of the luminescent markers used nowadays as well as fluorescent microscopes used for in vitro applications use blue excitation, which also can be used is our case. However, our further studies will be also devoted to sensitize Mn2+ emission through up-conversion by the co-doping with Yb3+ ions. Therefore we will be able to use NIR excitation. This aspect is commented in the revised version of our manuscript.
- The text describing the potential occurrence of (bidirectional) energy transfer between Mn and Eu is confusing and difficult to follow. The donor -acceptor scheme should be applied to both Mn and Eu with the two spectral evidences for a potential energy transfer (excitation spectra and the evolution of the emission decays of the donor (Mn, Eu) at fixed concentration with increasing concentration of the acceptor (Eu, Mn). Although both evidences are partially discussed by the authors, the text should be shortened and more systematised.
Author's replay:
Our main conclusion from performed studies is the fact that there is no interionic energy transfer between Mn2+ and Eu3+ ions. Performed measurements including excitation spectra and emission spectra at constant Mn2+ or Eu3+ concentration, as well as luminescence decay profile does exclude interionic energy transfer between different ions. Only energy diffusion among Mn2+ and (separately) Eu3+ ions occurs in our systems. However, these processes are difficult to experimentally confirm. Motivated by the suggestion of the Reviewer we have deceided to reconstruct the results and discussion part to make it more systematized.
- In case of Mn2+ , aliovalent doping induces defects via charge-compensation, (probably, oxygen vacancies or other type ?). Though authors discuss the appearance of the so-called "defect states" this is not equivalent with the generation of defects, whatever their nature. Please, give a short description of the defects generated whose amount is likely to be significant for 10% Mn2+. Their distribution as near-neighbour or/and next near- neighbour to Eu is expected to affect drastically the Eu luminescence (emission shape and dynamics).
Author's replay:
Thank you very much for this suggestion. We totally agree with this suggestion and this was commented in the revised version of our manuscript. Some other authors (OXYGEN VACANCY-TRANSITION METAL-ION IMPURITY ASSOCIATION IN SrTi03, Solid State Communications, Vo1.45,No.10, pp.903-906, 1983) observed similar behavior in the systems under investigations.
Minor comments:
- Some references are incomplete: e.g. [2 - 18, 21, 24 - 28, 31, 32, 34 - 39, 41 - 43, 45 - 52] (it lacks the title of the article, Journal, year, vol, issue and pages). Please, correct them.
Thank you very much for this comment. Changes have been made.
- Also, in Introduction, row 49 “non-invasive nanothermometers such as YAG garnets” should probably cite Ref 15 and 20 not22. In ref 22 the study is on LiLaP4O12 nanocrystals. 3. Further, the authors should cite Revised effective ionic radii and systematic studies of interatomic distances in halides and chalcogenides. Acta Crystallographica Section A: Crystal Physics, Diffraction, Theoretical and General Crystallography 1976, 32 (5), 751-767) when discussing the ionic radii of the cation/dopants. Apparently, the XRD patterns do not match closely those of PDF of LaPO4 (Figure 1a). Probably, it would be better of authors repeat the XRD measurements.
Author's replay:
Thank you for this comment. Suggested reference was cited in the revised version of our manuscript. Unfortunately we are unable to provide reference pattern for hexagonal form of GdPO4. Therefore, we used the one for hexagonal LaPO4 (ICSD 31564) According to our studies all diffraction peaks correspond to the reference data with small shift of our data toward higher 2theta. This effect is expected due to the difference between La3+ and Gd3+ ionic sizes.
- Further, the TEM images in Figure 1e displays some agglomeration contrary to the authors description. Please revise the text accordingly.
Author's replay:
Observed effect is related to the layering of the nanoparticles when applied to a carbon grid for TEM measurements and not to their aggregations. This was commented in the revised version of the manuscript. .
- Row 200 “narrow sharp band at 405 nm is observed”. Here the authors describe the excitation spectra monitoring the 550 nm Mn2+ emission although in the Figure2b it can be observed that the excitation spectra (the blue lined spectra) displays the absorption peak at ~450 nm. Please verify the peak wavelength.
Author's replay:
Thank you very much for this suggestion. There was some error in the displaying fo the data presented in Fig. 2. which has been corrected in the revised version of our manuscript. All obtained excitation spectra measured for Mn2+ emission (See also SI) reveal absorption band at 405 nm.
- What is the significance of the absorption peaks at 300 – 325 nm ? How the authors extract the contribution of the particular ion’s emission intensity from the total emission spectra (spectral deconvolution ?)
Author's replay:
Thank you very much for this comment. These bands are related to the Gd3+ absorption and this information is provided in the revised version of our manuscript. In order to provide temperature readout as simple as possible simply the integral emission intensity in given spectral range was considered without deconvolution. We are aware that in this case some part of the signal is missed due to a spectral overlap of the bands but due to the fact that f-f transitions of Eu3+ are characterized by the narrow bandwidth the missed intensity is relatively low.
- It appears that Table 1, is not cited or discussed in the manuscript.
Author's replay:
Thank you very much for this suggestion. Table 1. is now cited in the revised version of our manuscript.
- Please discus the thermometers performance in comparison with the reported literature. In Figure S4b, the asymmetry ratio value of the 0.5%Eu sample is out of trend. Can the authors explain this ? .
Author's replay:
Indeed the asymmetry ratio for 0.5% of Eu3+ is out of the observed trend and we are not sure what is the reason of this behavior. The variation of asymmetric ratio is not high (around 7% from the value obtained for 0.2% of Eu3+ and 1% of Eu3+ ions). Nevertheless this value does not change the general trend and this concentration of Eu3+ ions was not used in the further optimization process presented in our manuscript. As suggested by this Reviewer and other Reviewers the manuscript need to be shortened. Therefore, the comparison of the relative sensitivities obtained for other manganese based luminescent thermometer was given in a Table 1 and shortly commented. This comparison enables to conclude that presented in this manuscript GdPO4 nanocrystals are very promising candidates for luminescent temperature sensing.
- Please , /complete the caption of Figure S10 (temperature range). Further, Figure S8, the “Thermal evolution of integrated Eu3+” has an out of trend evolution for 0.5%Eu which needs some explanation. In Figure S9, the LIR evolution of 5% and 10% Eu is similar for the first two temperature values (30 and 32 °C), but the corresponding Sr are quite different (8.88%/°C versus ~4%/°C).
Author's replay:
Thank you very much for this comment. The caption of Figure S10 is completed in the revised version of our manuscript. In the case of Figure S8: the relatively low emission intensity of this sample and the especially Eu3+ bands lead to the fluctuation of the Eu3+ emission intensity as a function of temperature. Concerning Figure S9 it needs to be mentioned that in order to enable the Reader the comprehensive analysis LIRs were normalized to their initial value (at 30oC). The SR presented in figure b represents relative sensitivity which is dependent on the real value of LIR.
SR=(1/LIR)(dLIR)/(dT)*100%
This is the reason why there is a difference in SR between these samples. In the revised version of our manuscript the normalization fact was underlined in the Fig. 4 and Fig. S9.
- The authors should verify the LIR and Sr values. Some of the Figure and Figure captions in Supplemental Information are mislabelled with Mn3+ instead of Mn2+. Please correct the English grammar and language through the text (local ions symmetry should be replaced with ions local symmetry and so on). Also, several verbs are missing.
Author's replay:
Thank you for your suggestion. We believe that revised version our manuscript is acceptable.
Reviewer 2 Report
In my opinion, the work includes a very detailed and complete study of the synthesis, the cytotoxicity evaluation and the analysis of the optical properties of GdPO4 colloidal samples: codopated with Mn2 +, Eu3 + of application as high sensitivity luminescent nanothermometers that operate in The physiological temperature range. It is a very new and very complete work and I encourage the authors to continue researching their applications.
Author Response
In my opinion, the work includes a very detailed and complete study of the synthesis, the cytotoxicity evaluation and the analysis of the optical properties of GdPO4 colloidal samples: codopated with Mn2 +, Eu3 + of application as high sensitivity luminescent nanothermometers that operate in The physiological temperature range. It is a very new and very complete work and I encourage the authors to continue researching their applications.
Author's replay:
Thank you very much for your comments.
Reviewer 3 Report
The authors describe the synthesis and characterization of colloidal GdPO4: Mn2+, Eu3+ nanoparticles for non-contact temperature sensing for in vitro experiments. The topic is interesting for a wide audience of scientists in biomedicine as well as for nano-scientists.
The authors report a very detailed and robust characterization of their nanomaterials flanked by preliminary cytotoxicity experiments, the body of the results converge to indicate a specific composition GdPO4: Mn2+, Eu3+ as luminescent nanothermometer.
The manuscript deserves publication provided the authors complete an extensive English revision and organize and compact the results dividing the manuscript in sections.
Although the introduction is informative and well written the “Results and Discussion” section requires an important re-organization. It is very difficult to follow the plethora of experimental results reported in the main text and in the supplementary material. The section is a Joyce-type stream of consciousness not recommended for scientific literature.
The authors could also provide comments to some minor questions:
- The authors used relatively high temperature during the synthesis, e.g. “kept at 200°C for 2 hours”, did the authors consider also eco-sustainable methods such as micro-wave assisted synthesis ?
-Photoluminescence decays: did the authors use a specific gate or delay time? This information can be added in the experimental section together with comments on residual short-lived species emission.
The title can be shortened moving “synthesis, cytotoxicity assessment and optical properties characterization to the key-words.
Author Response
The authors describe the synthesis and characterization of colloidal GdPO4: Mn2+, Eu3+ nanoparticles for non-contact temperature sensing for in vitro experiments. The topic is interesting for a wide audience of scientists in biomedicine as well as for nano-scientists.
The authors report a very detailed and robust characterization of their nanomaterials flanked by preliminary cytotoxicity experiments, the body of the results converge to indicate a specific composition GdPO4: Mn2+, Eu3+ as luminescent nanothermometer.
The manuscript deserves publication provided the authors complete an extensive English revision and organize and compact the results dividing the manuscript in sections.
Although the introduction is informative and well written the “Results and Discussion” section requires an important re-organization. It is very difficult to follow the plethora of experimental results reported in the main text and in the supplementary material. The section is a Joyce-type stream of consciousness not recommended for scientific literature.
Author's replay:
Motivated by the suggestion of the Reviewer reorganization and correction are provided in the revised version of the manuscript. The Results and discussion section was divided in the subsections as suggested by the Reviewer.
The authors could also provide comments to some minor questions:
- The authors used relatively high temperature during the synthesis, e.g. “kept at 200°C for 2 hours”, did the authors consider also eco-sustainable methods such as micro-wave assisted synthesis ?
Author's replay:
Thank you very much for this suggestion however, the method of synthesis using water as a reaction medium, carried out in an autoclave, is classified as a green chemistry method. One of the most important advantage of this method is the easiness in the scaled up of the synthesis product. However, we did not consider the use of microwave assisted technique due to the limitation of the laboratory equipment.
-Photoluminescence decays: did the authors use a specific gate or delay time? This information can be added in the experimental section together with comments on residual short-lived species emission.
Author's replay:
Thank you for your comment. In our experiments we did not use delay time/specific gate because it could lead to the omission of some important information concerning the potential energy transfer between ions. Therefore this information was not provided in our manuscript.
The title can be shortened moving “synthesis, cytotoxicity assessment and optical properties characterization to the key-words.
Author's replay:
Thank you very much for your suggestion. However, if possible, we prefer to leave the title in its current form since it provide the Reader clear and simple information about the manuscript content. According to the suggestion of the Reviewer we additional keywords.
Round 2
Reviewer 1 Report
In the reply letter and resubmitted manuscript, the authors have addressed most of my concerns, therefore I recommend publication.
This manuscript is a resubmission of an earlier submission. The following is a list of the peer review reports and author responses from that submission.
Round 1
Reviewer 1 Report
Marciniak and coworkers report on the preparation and characterization of a series of colloidal GdPO4 nanocrystals doped with Eu3+ and Mn2+ions, which are proposed as luminescent thermometers in the temperature range of biological interest, and tested in vitro.
While the presented results may be of interest to a wide community, the work presents two main fundamental gaps.
First, the large amount of provided data are not clearly presented. On the one hand, many pieces of information are missing, e.g. when the % of Mn2+ are changed the % of Eu3+ is not given and viceversa (Figure S2 and S4) so that it is difficult to follow the discussion; on the other hand, it seems that in some cases information is wrong/misleading, e.g. is it true that sample in figure 4a is 15% of Mn2+, or is it rather 10%? Why does sample with 15% Mn2+ appear to show no dependence from graphs in fig 4b-c, and spectra in fig 4a and S4 seem to provide very different information?
In addition, it seems from figure S2 that only one sample (10% Mn2+,??% Eu3+) features a significant temperature dependence. Basically, that appears the only sample worth describing as a luminescence thermometer. Further, since only 10% Mn2+ provides an interesting response, why Eu3+ doping degree was changed with Mn2+ 15%, which is already observed to show no dependence on T? I suggest to tune Eu3+ concentration with Mn2+ 10% instead.
The second main gap of the manuscript is the lack of explanation behind the observed behaviour: why T dependence is only observed at 10% Mn2+ (with some irregular T dependence at other concentrations, which yet do not form any trend? Is the T dependence dominated by surface defects, local ion simmetry or other phenomena which are not controlled (and not directly related to) with dopant concentration?
Some related questions to these two main gaps are:
what is the Thermal dependence of emissions of samples with only Mn2+ or Eu3+? is it necessary to have both dopants in order to have a thermal dependence? or is it merely convenient for ratiometric response? Thermal trends in fig S4 seems to show largest variations for Mn2+ 15%, but these spectra are not shown in fig S2 and they seem to be in contrast with fig 4b-c. In addition, trends relative to 10% Mn2+ seem here to be the less promising, while 5% Mn2+may appear as the most promising for ratiometric LIR due to largest contrast between Mn2+ and Eu3+ emissions... this appears very confusing to the reader! Is the best thermal trend (apparently for Mn2+ 10% and Eu3+ 1%??) also reversible? can the temperature by cycled up and down? what is the LIR behaviour upon T cycling? lifetime analysis should be provided, at least for the best performing sample(s), also vs temperature, which may provide details on the photophysics behind thermal dependence of the emissions.
In conclusion, I believe the present manuscript is still too preliminary for publication. Additional experiments and additional data analysis should be performed, and the presentation of results needs substantial improvement.
Author Response
Reviewer 1
Marciniak and coworkers report on the preparation and characterization of a series of colloidal GdPO4 nanocrystals doped with Eu3+ and Mn2+ions, which are proposed as luminescent thermometers in the temperature range of biological interest, and tested in vitro. While the presented results may be of interest to a wide community, the work presents two main fundamental gaps.
First, the large amount of provided data are not clearly presented. On the one hand, many pieces of information are missing, e.g. when the % of Mn2+ are changed the % of Eu3+ is not given and viceversa (Figure S2 and S4) so that it is difficult to follow the discussion; on the other hand, it seems that in some cases information is wrong/misleading, e.g. is it true that sample in figure 4a is 15% of Mn2+, or is it rather 10%? Why does sample with 15% Mn2+ appear to show no dependence from graphs in fig 4b-c, and spectra in fig 4a and S4 seem to provide very different information?
Authors reply:
Thank you very much for this suggestion. We have corrected captions of all Figures and provided missed information. Reviewer has absolutely right that there was a mistake in the caption of Figure 4. In Fig. 4a the thermal evolution of GdPO4:10%Mn2+, 1% Eu3+ is presented. In the case of each dopant concentration some difference in the luminescence intensity ratio has been found which confirms that each dopant concentration can be used for noncontact temperature sensing. We have modified Figure S2 to make easier to recognize these changes. However in the case of some of the dopant concentration thermal changes of LIR provide low relative sensitivity. Therefore the aim of our studies was to optimize the dopant concentration in order to enhance the relative sensitivity of luminescent thermometers.
In addition, it seems from figure S2 that only one sample (10% Mn2+,??% Eu3+) features a significant temperature dependence. Basically, that appears the only sample worth describing as a luminescence thermometer. Further, since only 10% Mn2+ provides an interesting response, why Eu3+ doping degree was changed with Mn2+ 15%, which is already observed to show no dependence on T? I suggest to tune Eu3+ concentration with Mn2+ 10% instead.
Authors reply:
Thank you very much for your comment. Actually Eu3+ is expected to be insensitive to temperature changes. However, if the energy transfer between transition metal ion and lanthanide ion occurs the higher Eu3+ concentration should enhance thermal quenching of transition metal ions luminescence due to thermally activated energy transfer. Therefore we have decided to verify if this process occur in our case. This is why thermal evolution of emission spectra of nanoparticles with different Eu3+ concentration. In the course of our studies it has been found that high Mn2+ concentration modifies the local ions symmetry facilitating luminescence thermal quenching of Mn2+. The highest modification which was represented as the shortest lifetime of both Eu3+ and Mn2+ excited state was found for 15%Mn2+. Therefore we used this dopant concentration to determine the influence of the Eu3+ concentration on the spectroscopic properties of examined system. As it can be noticed in Fig. 4 e the higher concentration of Eu3+ the lower relative sensitivity of luminescent thermometer.
The second main gap of the manuscript is the lack of explanation behind the observed behaviour: why T dependence is only observed at 10% Mn2+ (with some irregular T dependence at other concentrations, which yet do not form any trend? Is the T dependence dominated by surface defects, local ion simmetry or other phenomena which are not controlled (and not directly related to) with dopant concentration?
Authors reply:
Our investigations revealed that mainly local ions symmetry affect thermal dependence of emission intensity of Mn2+ ions in the analyzed systems. The measurements of the kinetics of the excited state revealed that the increase of Mn2+ concentration leads to the shortening of the decay time of both Mn2+ and Eu3+ excited states with exactly the same dependence, while such dependence has not been found for higher Eu3+ concentration. This process enhance the relative sensitivity of the luminescent thermometer while the second one hinders thermal changes of LIR. However this effect lead also to the lowering of the absolute emission intensity. Unfortunately we are unable to eliminate one of them without modification the other one. However in the case of high concentration of Eu3+ ions the energy diffusion to the surface defect states may hinder the relative sensitivity. Therefore from this point of view the lower Eu3+ concentration the higher relative sensitivity. On the other hand low emission intensity observed for low Eu3+ content may significantly reduce the accuracy of temperature readout. Therefore 1% of Eu3+ concentration seems to be optimal since it meets both these requirements, namely high emission intensity and high relative sensitivity. These information has been clarified in the manuscript.
Some related questions to these two main gaps are:
what is the Thermal dependence of emissions of samples with only Mn2+ or Eu3+? is it necessary to have both dopants in order to have a thermal dependence? or is it merely convenient for ratiometric response?
Authors reply:
These samples has been synthesized and the dependence of their emission intensity as a function of temperature has been provided in the Supporting Information of the revised version of the manuscript. It can be clearly seen that the GdPO4:Eu3+ revealed high luminescence stability in the investigated temperature range. On the other hand for the sample doped only with 10%Mn2+ similar decrease of Mn2+ emission intensity to that one observed for 10%Mn2+, 1%Eu3+ was found. This result confirm that energy transfer between Mn2+ and Eu3+ does not occur in the synthesized nanocrystals. Therefore co-doping with Eu3+ does not enhance the relative sensitivity. The most important advantage from the co-doping is the fact that it enables ratiometric temperature sensing. These information are provided in the revised version of the manuscript.
Thermal trends in fig S4 seems to show largest variations for Mn2+ 15%, but these spectra are not shown in fig S2 and they seem to be in contrast with fig 4b-c. In addition, trends relative to 10% Mn2+ seem here to be the less promising, while 5% Mn2+may appear as the most promising for ratiometric LIR due to largest contrast between Mn2+ and Eu3+ emissions... this appears very confusing to the reader! Is the best thermal trend (apparently for Mn2+ 10% and Eu3+ 1%??) also reversible? can the temperature by cycled up and down? what is the LIR behaviour upon T cycling? lifetime analysis should be provided, at least for the best performing sample(s), also vs temperature, which may provide details on the photophysics behind thermal dependence of the emissions.
Authors reply:
Thank you very much for your valuable comment. There was a mistake in the legend of Fig S4 a and b. Corrected description of the curves presented in this Figure has provided in the revised version of our manuscript. We apologize for this mistake which indeed could be confusing to the reader. Additional measurements of the change of LIR for the GdPO4:10%Mn2+, 1%Eu3+ confirm reversibility of observed changes. These information has been shown in Supporting Information of the revised version of our manuscript.
Reviewer 2 Report
The scientific content of this ms is interesting and this work thus deserves – according to my opinion – acceptance in NANOMATERIALS. As a matter of fact this is the best ms I have ever reviewed for the journal. I am sure that the paper will attract the intense interest of scientists working in several areas of Nanotechnology, and especially those related to biomedicine. Also, I do believe that the article will receive a respectable number of citations in the future. Salient features of this work – which support my proposal for acceptance – are : (a) The authors have developed a smart strategy (involving precipitation and hydrothermal synthesis) to obtain GdPO4 : MnII, EuIII nanoparticles. Orthophosphates co-doped with transition-metal and 4f-metal ions had not been developed before. (b) The authors propose a sensitive (in the physiological range of temperature) luminescent nanothermometer based on their material; and (c) Cytotoxicity experiments confirmed biocompatibility of the GdPO4/15% MnII/1% EuIII material. Overall, this work shows that colloidal GdPO4 doped with MnII and EuIII is a promising candidate as luminescent nanothermometer for in vitro applications. The ms. is well written and nicely organized. The quality of figures is good and the references list covers the topic under study more than satisfactorily. The Supporting Info is valuable for the readers. Although I tried hard, I could not locate scientific mistakes/errors.
Based on the above mentioned, I am glad because I can propose acceptance of this fine work in NANOMATERIALS. Minor comments/suggestions to be taken into account by the authors:
(1) “Introduction” is rather long and could be condensed.
(2) Did the authors try to prepare and test analogous materials with other lanthanide(III) emitting ions, e.g. TbIII? A comment is necessary at this point.
(3) “Conclusions”: The perspectives of this work should be mentioned.
(4) The authors should add 2-3 sentences briefly describing the nature and properties of orthophosphates doped exclusively with lanthanide ions, namely to briefly analyze the content of ref. [30].
Author Response
Reviewer 2
The scientific content of this ms is interesting and this work thus deserves – according to my opinion – acceptance in NANOMATERIALS. As a matter of fact this is the best ms I have ever reviewed for the journal. I am sure that the paper will attract the intense interest of scientists working in several areas of Nanotechnology, and especially those related to biomedicine. Also, I do believe that the article will receive a respectable number of citations in the future. Salient features of this work – which support my proposal for acceptance – are : (a) The authors have developed a smart strategy (involving precipitation and hydrothermal synthesis) to obtain GdPO4 : MnII, EuIII nanoparticles. Orthophosphates co-doped with transition-metal and 4f-metal ions had not been developed before. (b) The authors propose a sensitive (in the physiological range of temperature) luminescent nanothermometer based on their material; and (c) Cytotoxicity experiments confirmed biocompatibility of the GdPO4/15% MnII/1% EuIII material. Overall, this work shows that colloidal GdPO4 doped with MnII and EuIII is a promising candidate as luminescent nanothermometer for in vitro applications. The ms. is well written and nicely organized. The quality of figures is good and the references list covers the topic under study more than satisfactorily. The Supporting Info is valuable for the readers. Although I tried hard, I could not locate scientific mistakes/errors.
Based on the above mentioned, I am glad because I can propose acceptance of this fine work in NANOMATERIALS. Minor comments/suggestions to be taken into account by the authors:
“Introduction” is rather long and could be condensed.
Authors reply:
We appreciate the suggestion of the Reviewer. In the Introduction section of our manuscript we tried to introduce the Reader in the field of luminescent thermometry of nanomaterials doped with transition metal ions in the most condensed form. We believe that elimination of the some of the data may lead to the misunderstanding of the data presented in the manuscript.
Did the authors try to prepare and test analogous materials with other lanthanide(III) emitting ions, e.g. TbIII? A comment is necessary at this point.
Authors reply:
Thank you very much for this suggestion. The main problem with the use of the transition metal and lanthanide co-doped systems for luminescent thermometry is associated with the two aspects. First of them is the fact that excitation wavelength should simultaneously and separately excite both of ions. The later one is associated with the spectral overlap of the emission of transition metal ions and lanthanide ions. Therefore we have decided to use Eu3+ ions which meet both of these requirements. However even in this case we were forced to use emission band of Eu3+ ion localized around 690 nm. In the case of Tb3+ ions the spectral overlap of their emission with the Mn2+ emission would reduce the accuracy of the noncontact temperature readout.
(3) “Conclusions”: The perspectives of this work should be mentioned.
Authors reply:
According to the suggestion of the Reviewer we have added following sentences to the Conclusions:
“Due to the high application potential of the obtained luminescence nanothermometer showing high relative sensitivity in the physiological temperature range, more advanced in vivo and in vitro tests are planned. Further modification of the host materials which enable lowering of the crystal field strength should enhance the relative sensitivity of luminescent thermometer.”
(4) The authors should add 2-3 sentences briefly describing the nature and properties of orthophosphates doped exclusively with lanthanide ions, namely to briefly analyze the content of ref. [30].
Authors reply:
According to the suggestion of the Reviewer we have added following text to the Introduction:
“The use of LaPO4 and YPO4 orthophosphates nanocrystals co-doped with lanthanide ions for noncontact temperature and pressure sensing has been reported by Runowski et al [30]. Provided versatile studies confirmed that these host materials enable determination of temperature with relative sensitivity as high as S=3.0%/oC for LaPO4:Yb3+,Tm3+ and 2.25% for YPO4:Yb3+,Tm3+ at 29oC. “
Round 2
Reviewer 1 Report
In the revised version authors did not satisfactorily face my criticisms.
In particular, while amendements to the captions partially solved the problem of clarity of presentation of data, the discussion was not significantly improved, many points in the presentation and discussion of data remain obscure, and my overall feeling is that it is very difficult to extract coherent information from the presented data.
In addition, a large part of the study was performed on a set of samples (15% Mn2+, increasing % of Eu3+) that has no interest, since the LIR is practically zero. My suggestion to perform the investigation on samples with 10% Mn2+ (which show the most interesting behaviour) and change the % of Eu3+ was not even mentioned in the authors reply. Sentences in the text such as "As it can be clearly seen for low Eu3+ concentration the reduction of the Mn2+ emission band can be observed in the analyzed temperature range.", related to figure S2, are an oversimplification of the observed behaviour: it is not true that in samples with low % Eu3+ the Mn2+ emission show relevant Thermal changes, for example see sample with 0.2% Eu3+ (fig S2b).
The sentence "Evidently for most of the Mn2+ concentration the SR decreases with increase of the temperature. Only curve for GdPO4: 15%Mn2+, 1%Eu3+ nanocrystals reveals opposite tendency." is a misleading discussion of the data, since SR for such sample is close to zero. This is the main observation, and this is not discussed at all, with the discussion pointing at non-relevant details.
Concerning the closing remark "Taking advantage from these studies it can be concluded that GdPO4: 1% Eu3+, 15% Mn2+ nanoparticles reveals the best performace for temperature sensing." I do not understand its meaning: the best performance was shown for the 10% Mn2+ 1% Eu3+ sample, while the 15% Mn2 1% Eu3+ sample show LIR close to zero.
Finally, errors contained in the first submitted version (with wrong sample names throughout captions and text) are very important and witness a too high level of approximation. I fear that similar errors are still present in the revised version, for example in the caption of figure S5.
Concerning errors in the presentation of data, I do not find any correlation between the data shown in fig 4e and in fig S6b, which should in principle contain the same data at 42°C.